# Portal Hemodynamics after Living-Donor Liver Transplantation: Management for Optimal Graft and Patient Outcomes—A Narrative Review

**Kishore GS Bharathy** [1,*] and **Sunil Shenvi** [2,3]

1    Department of HPB Surgery and Liver Transplantation, Fortis Hospital, Bangalore 560076, India
2    Department of GI, HPB and Multiorgan Transplantation, Trustwell Hospitals, Bangalore 560002, India; sunilshenvi@gmail.com
3    Department of GI & HPB Surgery, Jayadev Memorial Rashtrotthana Hospital, Bangalore 560098, India
*    Correspondence: kishoregsb@gmail.com; Tel.: +91-9540946806

**Abstract:** Background: When a partial liver graft is transplanted into a recipient with portal hypertension, it is subject to sinusoidal shear stress, which, in good measure, is essential for regeneration. However, portal hyperperfusion which exceeds the capacity of the graft results in the small-for-size syndrome manifested by ascites, cholestasis and coagulopathy. This review discusses intraoperative hemodynamic variables that have been described in the literature, and inflow modulation strategies and their outcomes. Apart from using donor grafts which are of adequate size for the recipient weight, portal hemodynamics are an important consideration to prevent early allograft dysfunction, graft failure and mortality. Summary: Understanding normal portal hemodynamics, how they change with the progression of cirrhosis, portal hypertension and changes after the implantation of a partial liver graft is key to managing patients with living-donor liver transplantation. If the intraoperative measurement of portal flow or pressure suggests graft portal hyperperfusion, inflow modulation strategies can be adopted. Splenic artery ligation, splenectomy and hemiportocaval shunts are well described in the literature. The proper selection of a donor to match the recipient's anatomic, metabolic and hemodynamic environment and deciding which modulation strategy to use in which patient is an exercise in sound clinical judgement. Key message: The intraoperative assessment of portal hemodynamics in living-donor liver transplant should be standard practice. Inflow modulation in properly selected patients offers a point-of-care solution to alter portal inflow to the graft with a view to improve recipient outcomes. In patients with small (anatomically/metabolically) grafts, using inflow modulation can result in outcomes equivalent to those in patients in whom larger grafts are used.

**Keywords:** portal hemodynamics; living-donor liver transplantation; small-for-size syndrome; inflow modulation; splenic artery ligation; splenectomy

## 1. Introduction

A successful living-donor liver transplantation (LDLT) requires the new graft to adapt and regenerate rapidly in a new hemodynamic milieu where it is subject to increased portal flow from a dilated portomesenteric vascular bed. The liver has to be of a good quality and have sufficient parenchymal volume to accommodate this portal flow and meet the metabolic demands of a compromised recipient. As graft function settles, jaundice clears rapidly, ascitic output decreases and the overall wellbeing of the patient improves. The selection of the best possible donors and the optimization of recipients are preoperative variables, while vigilant postoperative care is essential if patients are to do well. Intraoperative measurements of hemodynamic variables offer a 'point of care' solution to manage portal perfusion to the graft liver by inflow modulation. This review provides an overview of hemodynamics of the portal system and attempts to clarify the role of graft inflow modulation (GIM) strategies as relevant to LDLT.

## 2. Normal Splanchnic Hemodynamics

The liver is uniquely placed to receive blood from the gastrointestinal tract through the portomesenteric system and is a powerhouse of molecular metabolic activity acting through this gut liver axis. From a purely mechanistic point of view, it serves as a reservoir that returns blood from the abdominal organs to the heart. The liver receives around 1/4th of the cardiac output while constituting only 2.5% of the body weight [1], which amounts to 100–130 mL/min per 100 g of liver weight [2]. The portal vein delivers around 70–75% of the blood, carrying 50–70% of the oxygen requirement, while the hepatic artery supplies the rest [3,4]. Liver sinusoids hold 60% of the blood, while the capacitance vessels (hepatic artery, portal vein and hepatic veins) account for the remaining 40% [2]. Portal flow depends on the resistance offered by the liver bed as well as splanchnic and mesenteric arteriolar vascular tone. The lack of valves in the portal system helps maintain low pressure and low resistance. The normal range of portal venous pressure (PVP) ranges from 5 to 10 mmHg [5]. As blood flows through the liver towards the heart, the pressure head drops progressively. Sinusoidal pressure or hepatic venous wedge pressure (3 to 10 mmHg) is between the PVP (5–10 mmHg) and inferior vena cava pressure (1–2 mmHg) [6]. Figure 1 depicts the normal portal systemic circulation.

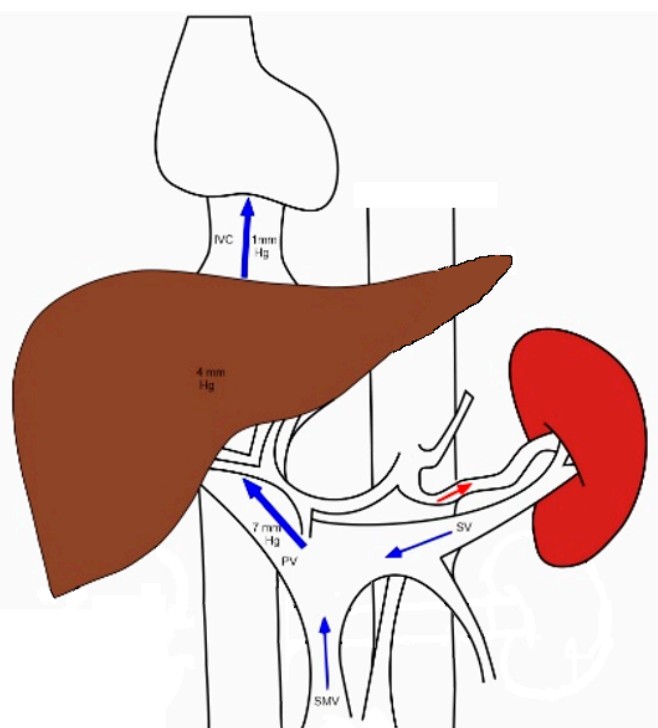

**Figure 1.** The major inflow to the liver comes from the portal vein, which receives blood from the spleen and the intestines via the splenic and mesenteric veins, respectively. The portal blood passes through a falling pressure gradient across the liver and reaches the systemic circulation via the hepatic veins into the right atrium. SMV—superior mesenteric vein; SV—splenic vein; PV—portal vein; IVC—inferior vena cava.

## 3. Changes That Occur in Splanchnic and Systemic Circulation in Chronic Liver Disease with Portal Hypertension

Portal hypertension is defined as a sustained mean pressure greater than 12 mmHg in the portal vein and its collaterals, which constitutes an increased risk for variceal bleeding and other complications [7]. Clinically significant portal hypertension is defined as hepatic venous pressure gradient (HVPG) > 10 mmHg. This is associated with a significantly higher risk of decompensation and mortality [8]. With the progression of liver parenchymal disease, the resistance to hepatopetal portal blood flow increases. The stellate cells lose

their normal orientation and morphologically turn into myofibroblast-like cells [9], and the sinusoidal endothelial cells become capillarized [10], thereby leading to the disappearance of the sieve-plate structure. Multiple collateral channels open up and divert blood away from the liver. (Figure 2) These portosystemic shunts result in splanchnic vasodilatation, and a hyperdynamic state ensues with high cardiac output and low systemic vascular resistance. Sodium and water retention occurs in response to this, with the expansion of plasma volume [11,12]. The liver and spleen, being solid organs, serve as compliance reservoirs while regulating mesenteric blood flow into the heart. There is a reciprocal relationship between hepatic and splenic sizes/volumes as blood flow is redistributed through portosystemic shunts.

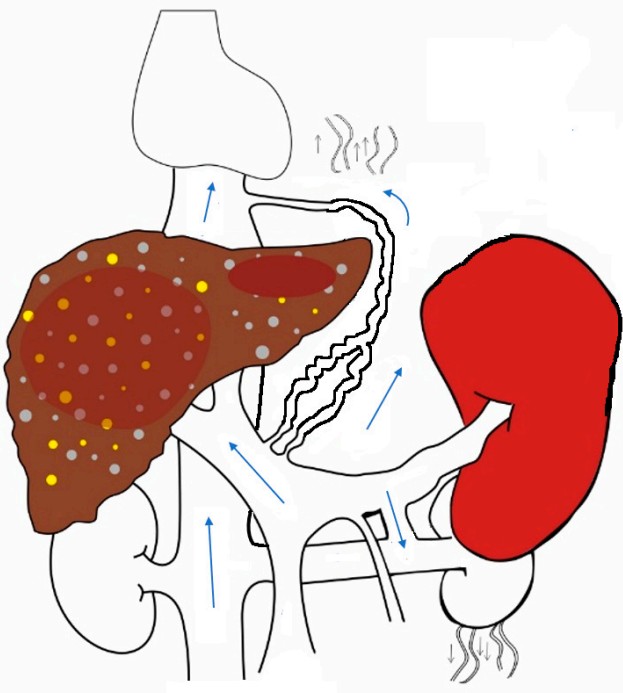

**Figure 2.** In liver cirrhosis, sinusoidal resistance to portal flow increases and portal pressure increases, resulting in splenomegaly and the opening up of multiple portosystemic collaterals. Portomesenteric blood bypasses the liver and reaches the systemic circulation through these abnormal channels.

## 4. The Hepatic-Artery-Buffer Response

The relationship between the portal and arterial blood flow to the liver is regulated by local paracrine mechanisms by adenosine. This is known as the hepatic-artery-buffer response (HABR), which was first described by Lautt [13]. Whenever portal flow increases, there is a corresponding decrease in hepatic artery flow. Adenosine, a vasodilator, is washed away from the sinusoids, and this leads to vasoconstriction and a decrease in arterial flow. Portal flow does not change reciprocally with changes in arterial flow. Portal flow in normal individuals is around 1500 mL/min, arterial flow is 300–400 mL/min and the ratio of portal to arterial flow is 2.5–3.5. With portal hypertension, in addition to the increase in portal flow, a decrease in arterial flow is also believed to contribute to graft dysfunction and ischemia to cholangiocytes. Post transplantation, partial liver grafts receive less hepatic artery flow compared to full grafts in absolute terms. However, the median hepatic arterial blood flow to the graft per 100 g of liver is not different in full or partial grafts. The ratio of portal-to-hepatic arterial flow increases from 6.6 to 15.4 post reperfusion in full and partial grafts, respectively [14].

## 5. Changes That Occur When a New Liver Is Transplanted into the Hyperdynamic Circuit

The relationship between pressure and flow is governed by the following equation. Portal Flow = Pressure gradient/Resistance. Resistance comes from the graft (both size and quality); flow and pressure are determined by the size of portal vein and extent of collateralization. The net flow per unit weight of liver depends upon the patency and the diameter of the portal vein, which is, in turn, inversely proportional to the extent and size of the portosystemic shunts. A sudden change in the pathologically altered hemodynamics by replacing the cirrhotic liver with a pliable donor liver results in a low-resistance pathway for the influx of a large amount of blood from the splenic vein (Figure 3). The compliance of the graft liver is higher than that of the cirrhotic liver, and, hence, resistance to the portal flow is much lower. Therefore, the volume of blood flowing through a unit gram of liver tissue is higher. The shear stress of the blood flowing through the sinusoids is the trigger for hepatic regeneration. Periportal hepatocytes come in contact with heptotrophic growth factors with an increase in sinusoidal permeability [15]. The portal flow should be optimal; too little flow hampers regeneration and graft function, and too much flow results in the all-too-well-known small-for-size syndrome (SFSS). Table 1 summarizes the many definitions of SFSS which have been proposed by different groups [16–21]. In 2015, Dahm et al. [18] proposed a definition of SFS dysfunction in a small partial liver graft (GRWR < 0.8) as the presence of two criteria (bilirubin > 100 umol/L, INR > 2, grade 3/4 encephalopathy) on three consecutive days in the first postoperative week after the exclusion of technical (arterial/portal occlusion, outflow congestion, bile leak), immunological (rejection) or infectious (cholangitis, sepsis) causes. Subsequently Hernandez-Alejandro et al. proposed a more comprehensive definition including portal flow (>250 mL/min/100 g liver) as a prerequisite in addition to size (GRWR < 0.8) [20]. They considered SFSS to be present if two of the four parameters (ascites, hyperbilirubinemia, prolonged INR, hepatic encephalopathy) were present (Table 1) in the absence of technical/immunological or infectious causes. The inclusion of portal flow marks the shift in understanding from size alone to size and flow paradigm in the pathogenesis of SFSS.

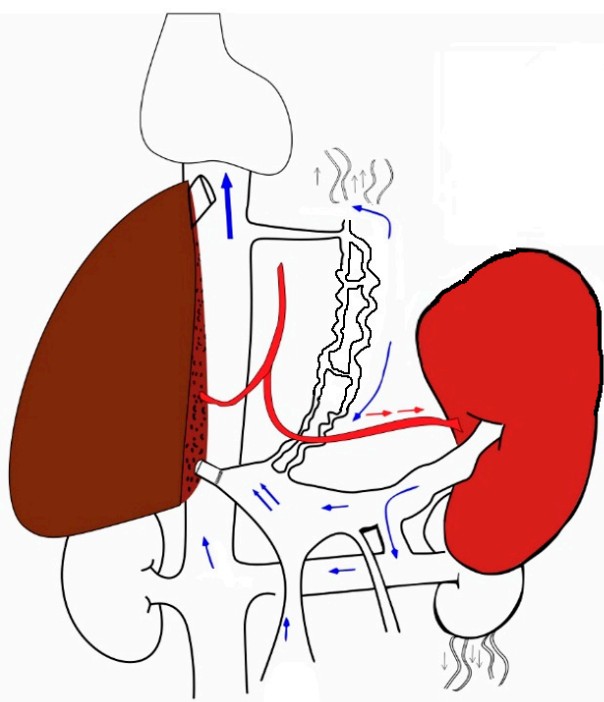

**Figure 3.** Depicts a right-hemiliver graft transplanted into a hyperdynamic circuit due to portal hypertension. The partial graft receives blood from the enlarged spleen.

**Table 1.** Definitions for small-for-size syndrome after liver transplantation.

| Author | Year | Terminology | Criteria |
|---|---|---|---|
| Soejima Y et al. [16] | 2003 | SFSS | Total Bilirubin > 5 mg/dL (on POD 14), ascites > 1 L (on POD 14) or ascites > 0.5 L (on POD 28) |
| Soejima Y et al. [17] | 2006 | SFSS | Total Bilirubin > 10 mg/dL (on POD 14), ascites > 1 L (on POD14) or ascites > 0.5 L (on POD 28) |
| Dahm F et al. [18] | 2006 | SFSD | Total Bilirubin > 100 µmol/L, INR > 2, Encephalopathy grade III-IV |
| | | SFNF | Retransplantation or death within the first postoperative week |
| Olthoff KM et al. [19] | 2014 | EAD | Total Bilirubin > 10 mg/dL (on POD 7) or INR > 1.6 (on POD 7) |
| Hernandez-Alejandro R et al. [20] | 2019 | SFSS | 1.    Size and flow prerequisite: GRWR < 0.8 + PVF > 250 mL/min/100 g<br>2.    Any two of the following:<br>Total Bilirubin > 5 mg/dL (lasts for 3 consecutive days within the first postoperative week or on POD 14)<br>INR > 2 (lasts for 3 consecutive days within the first postoperative week)<br>Ascites > 1 L (lasts for 3 consecutive days within the first postoperative week or on POD 14) or ascites > 0.5 L (POD 28)<br>Encephalopathy grade III–IV<br>3.    Absence of technical, infectious or immunological causes |
| Iesari S et al. [21] | 2019 | SFSS | Total Bilirubin > 20 mg/dL (lasts for 7 consecutive days after POD 7)<br>INR > 2 (lasts for 3 consecutive days within the first postoperative week)<br>Ascites > 1 L (lasts for 3 consecutive days within the first postoperative week or on POD 14) or ascites > 0.5 L (POD 28)<br>Encephalopathy grade III–IV |

SFSS—small-for-size syndrome; SFSD—small-for-size dysfunction; SFNF—small-for-size nonfunction; EAD—early allograft dysfunction; POD—postoperative day; INR—international normalized ratio.

Optimal portal flow is critical for liver regeneration. In a study of 64 recipients of right-hemiliver grafts who (all with a graft-recipient-weight ratio [GRWR] > 0.8) had an uneventful postoperative course, patients with an initial portal venous pressure of 23 mmHg and postreperfusion pressure of 15 mmHg had the best regeneration after 3 months of transplantation [22]. The portal flows positively correlated with hepatic regeneration 2 weeks after transplantation [23]. Portal venous velocity in the early post-transplant period is an important factor in liver regeneration [24]. Valdecasas et al. showed that portal flow in the recipient increased almost four-fold one hour after reperfusion; this was, however, associated with no adverse events, as all the 22 recipients in this series received right-hemiliver grafts with a median GRWR of more than 1 [25]. Portal venous flow is an important determinant of patient and graft outcomes. If the portal venous flow post reperfusion is more than four times the flow rate observed in donors (360 mL/min per 100 g), it is predictive of graft failure; flow rates less than half of that found in donors that resulted in poor survival [14]. An elegantly conducted hemodynamic study in 28 recipients with cirrhosis who underwent orthotopic liver transplantation (OLT) evaluated parameters before transplant and after six monthly intervals for a mean follow-up period of 17 months. After OLT, most systemic hemodynamic parameters such as heart rate, mean arterial pressure, peripheral vascular resistance and cardiac index normalized. However, although spleen size decreased it continued to be larger than in controls and did not return to normal [26].

A critical mass of liver which adapts to the new hemodynamic milieu and successfully hypertrophies to match the metabolic demand of the recipient is a sine qua non for good outcomes. A resetting of the flow and pressure systems occurs while the graft copes and functions in the new setting. The proper selection of donors, preoperative recipient evaluation, sound surgical technique, intraoperative management of recipient portal flow

and pressures and diligent postoperative care contribute towards this. These factors are briefly discussed, while intraoperative portal hemodynamics is the main focus of this paper.

## 6. Donor Selection

In an ideal world, there should be no shortage of cadaveric organs. However, in most Asian countries, LDLT is the predominant type of transplantation. In LDLT, the graft is a personal gift and, therefore, there is no competition with other recipients; issues of equity and justice do not arise. While matching donor–recipient pairs, all LDLT programs try to find double equipoise where donor risk is minimal and recipient benefit is maximal. Choosing the type of graft is an important decision in this context. For adult LDLT, hemiliver grafts are expected to provide a GRWR between 0.8 and 1. Donor safety remains the foremost concern while deciding whether to choose a right- or left-sided graft. A left-sided graft provides a safe volume of remnant in the donor. A future liver remnant of >35% in a donor is considered safe when a right-hemiliver graft is chosen [27]. Younger donors with minimal steatosis should be the norm in LDLT, as the quality of the liver is high, and this results in optimal regeneration of both the remnant and the graft.

## 7. Preoperative Recipient Evaluation

Recipient evaluation in chronic liver disease traditionally involves an assessment of the model for end stage liver disease (MELD) score as a surrogate marker for the extent of decompensation of the liver; the higher the MELD score, the sicker the recipient and the higher the metabolic demand. In addition to this, the performance status and functional reserve of other organs are evaluated. A preoperative assessment and discussion of the extent of portal hypertension should also be part of routine planning prior to LDLT. Spleen size, extent, distribution of portosystemic collaterals and status of the portal vein (diameter, presence of any thrombosis) are important preoperative parameters that can predict intraoperative portal hemodynamics. These are discussed subsequently. Portal hyperperfusion can be anticipated in the presence of a large spleen, extensive collateralization and a patent, dilated portal vein.

When matching donor–recipient pairs, the important factors to be considered are depicted in Figure 4. The two important decisions that are taken in this context are 1. right- vs. left-hemiliver grafts and 2. the need for inflow modulation—its type and timing. A left-hemiliver graft with inflow modulation has emerged as an equivalent alternative to a right-hemiliver graft. This offers a higher margin of safety to donors, while recipient outcomes are similar. Although left-sided grafts can be small for size, with hemodynamic modulation, SFSS does not manifest. This is especially true if the metabolic demand of the recipient is not high.

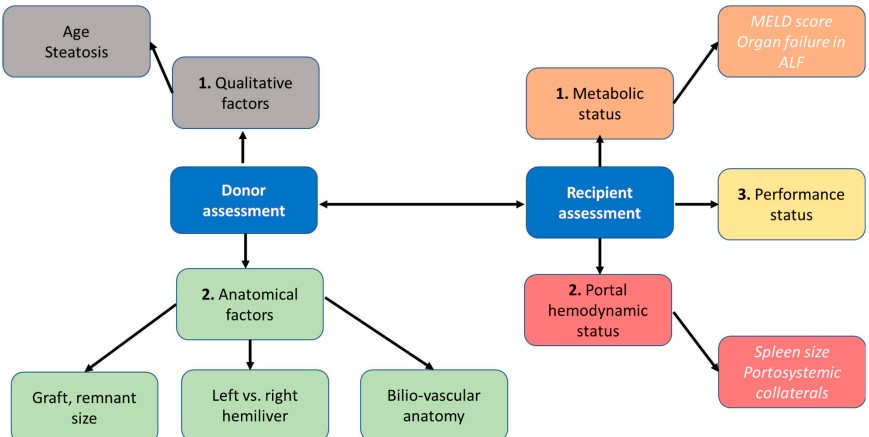

**Figure 4.** Factors to consider while matching donor–recipient pairs in living-donor liver transplantation. MELD—Model for End-Stage Liver Disease; ALF—Acute Liver Failure.

## 8. Outflow Reconstruction

In the presence of portal hyperperfusion, a compromised outflow compounds the effects of SFSS. Increased postreperfusion portal pressure in a normal-sized graft points towards a less-than-optimal outflow. All definitions of SFSS require the exclusion of technical problems such as outflow obstruction. If the hepatic venous outflow is wide and without angulation or twists, the volume of portal blood coursing through a unit gram of liver per min exits faster and, therefore, causes less shear-stress-related damage in the presence of the hyperperfusion of a smaller graft. Many LDLT centers take a small cuff of donor IVC (without causing any narrowing) to ensure a good outflow in right-hemiliver grafts. Ligating the phrenic vein provides extra length for safe clamp application. The Asan group used a fence graft of the recipient great saphenous vein to augment outflow [28]. Generally, the recipient cavotomy is larger than the donor hepatic vein diameter to ensure good outflow. Fujiki et al., in their insightful paper, emphasized the importance of outflow reconstruction in LDLT [29]. Their technique of venoplasty by uniting all the three hepatic-vein orifices in the recipient for the implantation of a left-hemiliver graft with caudate is novel and has resulted in excellent outcomes. They also used splenectomy for inflow modulation in these cases.

## 9. The Concept of Small for Size, Small for Flow

The paradigm of SFSS has evolved to encompass postreperfusion portal flow in addition to graft size. In reality, the two are closely interconnected. Using a smaller graft results in hyperperfusion per unit gram of liver tissue, assuming technical factors such as outflow and inflow reconstruction are perfect. The graft liver can be visualized as a bioengine capable of reversing the metabolic derangements associated with liver disease in the recipient. The graft size and quality are important considerations, especially if the metabolic demands are higher, for instance, in a sick recipient with high MELD score or in acute liver failure (ALF) with organ dysfunction. The exact threshold size of the graft which is the tipping point for graft failure after LDLT is unknown. It is difficult to predict due to a myriad combination of recipient metabolic and hemodynamic parameters. In liver resection, these thresholds for future liver remnant are better defined, below which posthepatectomy liver failure occurs. If a smaller graft is used in LDLT, portal hyperperfusion can be anticipated. Traditionally, a GRWR of less than 0.8 is considered a small-for-size graft. It is believed that a graft heavier than this threshold is able to meet the needs of the recipient. However, not all patients with a GRWR less than 0.8 develop SFSS. Conversely, grafts larger than this can also manifest SFSS. Many series have consistently used a GRWR up to 0.7 in low-risk patients, such as those with hepatocellular carcinoma (HCC) and a low MELD score, and achieved comparable outcomes to those with larger grafts. A Korean group evaluated 317 patients who underwent LDLT with right-hemiliver grafts over 7 years [30]. Of these, 23 had a GRWR < 0.7, 27 had a GRWR between 0.7 and 0.8 and 267 had a GRWR > 0.8. SFSS was higher in the first group (13% vs. 3.7% vs. 0.7%). Hepatic-artery thrombosis (HAT) was higher in group 1 (8.7% vs. 3.7% vs. 1.9%), which could be postulated to be due to the higher portal flows. However, there was no difference in the graft survival rates in the three groups. Reasons for larger grafts manifesting SFSS could be sick recipients with high MELD scores, anterior-sector congestion in the graft livers or a slightly suboptimal graft (older donor > 45 years or fat in the liver). Asenico et al. hypothesized that portal flow through the graft rather than just graft size contributes to hyperperfusion-related injury [31]. This paved the way for a wider adoption of intraoperative inflow modulation to prevent the manifestation of SFSS. Grafts with a GRWR < 0.6 have also been used successfully with the modulation of portal flow, resulting in acceptable patient and graft survival [32]. The Kyoto group pioneered the use of the a left hemiliver graft with caudate lobe. This enhances donor safety, as there is an adequate remnant right hemiliver. The graft is taken with the middle hepatic vein and the common MHV-LHV orifice results in good outflow. Usually for adults, the GRWR using left-hemiliver grafts is less than 0.8. Japanese groups from Kyoto and Kyushu advocate for splenectomy to reduce portal inflow

to the graft [33,34]. Their graft and patient survival rates were similar to those with a GRWR > 0.8. In acute liver failure, there is no long-standing portal hypertension; therefore, the size of the graft need not be large from a hemodynamic perspective, as portal inflow will not be huge. However, there is a cytokine storm that is usually brewing and a good-quality liver with sufficient mass (GRWR > 0.8) may be able to withstand the critical postoperative period better. While patients with fulminant liver failure may not have significant portal hypertension, they may still require a higher parenchymal mass to meet the metabolic demand of a sick recipient. Traditionally, a ratio of graft weight to standard liver volume (SLV) of recipient greater than 35–40% is considered adequate, as per most authors.

## 10. Recipient Portal Hemodynamic Status

### 10.1. The Role of the Spleen

The liver and spleen can be conceptually thought of as two solid organs that take the impact of the pressure of the portomesenteric flow and regulate it as it enters the systemic circulation into the right side of the heart. When the liver becomes stiff due to excess portal flow, its counterpart below the other side of the diaphragm enlarges to buffer the pressure dynamics in the system. Notwithstanding, the circulation of a large volume of blood through the enlarged portosystemic collaterals results in a hyperdynamic state with increased cardiac output. The spleen size is a direct reflection of the extent of portal hypertension. In extrahepatic portal-vein obstruction where the portal vein is replaced by a cavernoma, the spleen is massively enlarged with accompanying large spontaneous splenorenal shunts, retroperitoneal collaterals and gastroesophageal varices. A similar situation occurs in cirrhotic livers with an atretic portal vein or portal-vein thrombosis due to long-standing diminished hepatopetal portal flow. There have been many recent studies emerging on the importance of calculating the graft-weight-to-recipient-spleen-volume ratio (GSVR). The spleen-volume-to-graft-volume ratio (SVGVR) was shown to correlate with postreperfusion portal pressure irrespective of whether the graft was left or right [35]. SVGVR > 0.95 predicted the development of high portal pressure > 20 mmHg post reperfusion [35]. In a retrospective study, 246 recipients who underwent LDLT were divided into two groups based on GSVR (1.03 g/mL), with poorer survival in the low GSVR group [36]. Singh A et al. proposed a hyperperfusion index (HPi), a ratio of pressure gradient across the liver (postreperfusion PVP-central venous pressure) to the GSVR to as a composite index that predicts early allograft dysfunction (EAD) and mortality. Receiver-operating-characteristic (ROC) curves demonstrated cut-off values of HPi of 9.97 (EAD) and 16.25 (mortality) for these two outcome parameters, respectively [37].

### 10.2. Portosystemic Collaterals

The Asan group proposed a grading system for portosystemic collaterals based on preoperative computed-tomography (CT) evaluation. The size of these collaterals reflects the extent of portal hypertension. Varices were subjectively graded according to their diameter and tortuosity (Grade 1 is < 1 cm; grade 2 is 1–2 cm without tortuosity; grade 3 is >2 cm or tortuous). In 71 recipients of right-hemiliver grafts, varix score and GRWR significantly correlated with portal-flow velocity [38]. A detailed study on reconstructed CT portograms and intraoperative cineportograms can help in the decision to ligate these and arrest portal steal to facilitate prograde portal flow. The ligation of portosystemic shunts has been reported to result in better patient and graft outcomes [39]. In a prospective audit of 66 consecutive patients, shunt ligation was decided by a clamp test in the anhepatic phase; if the portal flow improved significantly after temporary occlusion of the shunt, it was ligated after reperfusion. Intraoperative-pressure monitoring was not performed in this study. If there are multiple lienorenal collaterals, the left renal vein can be ligated (Figure 5a). Test clamping of large collaterals after reperfusion can be conducted intraoperatively to assess its effect on portal flow and pressure if there is a decision to modulate portal flow (Figure 5b).

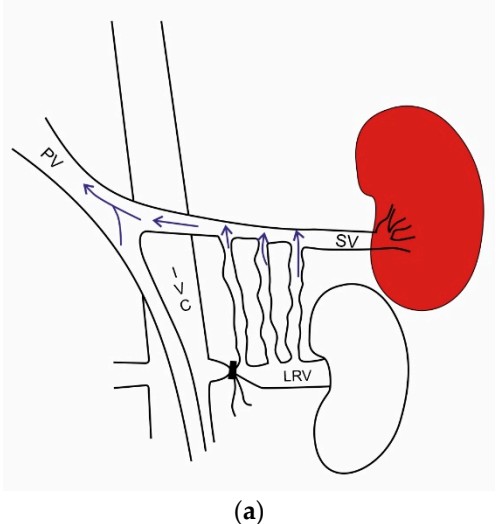 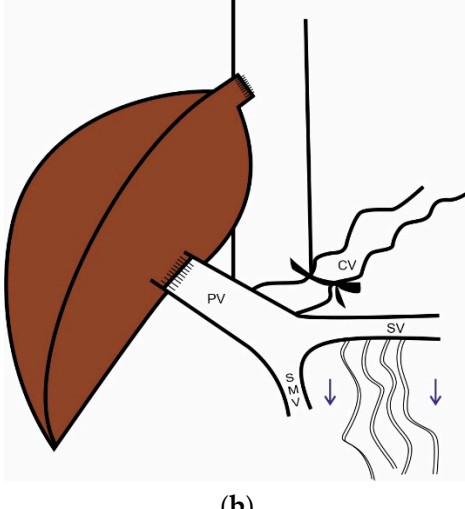

(**a**)                            (**b**)

**Figure 5.** (**a**) In the presence of multiple spontaneous lineorenal shunts, ligation of the left renal vein (LRV) close to the inferior vena cava closes these portosystemic channels and increases hepatopetal portal flow. (**b**) depicts the option of test clamping and ligation of the coronary vein (CV) and splenorenal collaterals for intraoperative modulation of portal flow.

## 11. Intraoperative Measurement of Portal Hemodynamics

Measuring portal blood flow and pressure in the recipient at the beginning of surgery and after reperfusion can help in intraoperative decision-making for inflow modulation. The technique of measurement has been described well in Japanese papers [40]. A catheter is placed into the portal vein through the inferior mesenteric vein. Continuous recordings can be obtained intraoperatively and in the postoperative period. This can be safely performed without increased risk of bleeding or infection. Direct cannulation of the portal vein for the measurement of pressure can also be conducted. However, it is subject to pressure variations and hemodynamics at that point in time. The best single time point to perform portal hemodynamic measurements is a few minutes after reperfusion of the graft. This provides a decision point to perform inflow modulation. If factors predictive of SFSS are present, inflow modulation (splenectomy) can be performed prior to recipient hepatectomy. Hemodynamic measurements in these cases can be performed at three time points (before modulation, after modulation and after reperfusion). Some authors have looked at portal pressure minus the central venous pressure and found it to correlate with outcomes. The measurement of HVPG which directly reflects sinusoidal pressure is another option; however, it is more invasive and not routinely practiced. It is recommended that portal pressure be measured in the recipient portal vein around 1–3 cm proximal to the anastomosis. Measurement beyond the anastomosis in the donor portal vein may not reflect true pressure, even after angle correction, as there is turbulent flow which may accentuated by any size discrepancy between donor and recipient portal veins [38]. While portal pressure is considered the cardinal marker for graft hyperperfusion by many groups, many authors emphasize the importance of portal flow per 100 g of liver tissue. The Kyoto group recommends inflow modulation if the postreperfusion portal pressure is more than 15 mmHg in older/ABO-incompatible donors [33]. The Kyushu group considers portal pressure more than 20 mmHg an indication for splenectomy [34]. Various thresholds of portal flow have been shown to predict SFSS development by different groups; Shimamura et al.—>260 mL/min per 100 g [41], Troisi et al.—>250 mL/min per 100 g [42]. In a series of 450 patients who underwent LDLT, there were 54 with a GRWR less than 0.8. A total of 6 out of these 54 developed SFSS. On multivariate analysis, only portal vein flow was responsible for the development of graft dysfunction. The GRWR and GRWR-to-SLV ratio did not predict SFSS or graft dysfunction. An ROC analysis showed a portal vein flow of 190 mL/min/100 g of liver to predict graft dysfunction with an area under ROC of 0.74 [43].

The relationship between postreperfusion portal pressure and flow is not linear [14]. Some patients with high portal pressures may have low portal flow, so modulation based on only pressure may lead to further decreases in portal flow with graft dysfunction.

Hepatic arterial flow also influences graft outcomes. If hepatic artery flow is less than 100 mL/100 g/min, it is a predictor of poor outcomes. Hepatic-artery thrombosis (HAT) or stenosis can lead to absent or reduced flows. A very high portal flow by virtue of the HABR can lead to reduced arterial flows in states of portal hyperperfusion and SFSS. In the setting of cholangitis or rejection as a second hit where sinusoidal resistance increases, HAT can occur. Low-flow states in the hepatic artery can result in poor oxygenation of cholangiocytes and contribute to the development of biliary strictures. Matsushima et al. retrospectively analyzed 1001 patients who underwent deceased-donor liver transplantation over a 10-year period [44]. Patients with high portal flows (>155 mL/min/100 g) had low HAF and a higher incidence of HAT and biliary complications. Compliance is calculated as the ratio of portal venous flow to portal pressure. Peak bilirubin and INR levels were better in more compliant grafts [45]. Surprisingly, not many studies have not evaluated this seemingly important composite parameter. Feng et al. have proposed an algorithm based on utilizing all available intraoperative hemodynamic parameters to decide on corrective measures/utilizing inflow modulation [4].

## 12. Inflow Modulation

The risks of SFSS while using SFS grafts can be mitigated with graft-inflow modulation (GIM). Graft survival of LDLT using SFSG has been shown to be equivalent to that of normal-sized grafts if GIM is performed [46]. In a retrospective analysis of 319 patients who underwent LDLT from 2007 to 2106, 189 did not require any modulation, 92 underwent successful modulation and 38 underwent modulation without reduction in portal pressures [33]. Modulation was performed mainly by splenectomy with an aim to reduce portal pressures below 15 mmHg. Patients whose portal pressures decreased with splenectomy had similar outcomes to those with who did not need modulation. In patients in whom splenectomy failed to decrease portal pressure, a donor age >45 years and ABO incompatibility were significant factors that predicted graft loss. Portal pressure >15 mmHg contributed to higher mortality in this subgroup of patients [33]. A recent systematic review by an expert panel strongly recommended GIM in SFS grafts, as there is [47] moderate evidence for enhanced recovery, although the evidence for reduction in mortality is low. Another recent metanalysis showed that GIM contributes to improved survival and decreased rates of SFSS [48].

### 12.1. Splenic-Artery Ligation (SAL)

SAL is perhaps the simplest intraoperative technique for inflow modulation. Troisi et al. proposed splenic-artery ligation in LDLT as a method to mitigate SFSS as early as 2003 [49]. SAL essentially cuts off the major source of arterial blood entering the spleen, thereby diminishing blood returning through the splenic vein into the portal vein (Figure 6a). The lesser sac is entered and the splenic artery is identified and looped over the superior border of the pancreas. If portal pressure is high after reperfusion, a bulldog clamp can be applied over the artery. If the pressure decreases to a threshold value less than 15- or 20-mmHg (as per institutional preference), the splenic artery is ligated. The proximal splenic artery is chosen for ligation so that the spleen continues to receive some blood supply from the short retro-pancreatic arteries. Ito et al. from Kyoto university demonstrated as early as 2003 that portal pressure in 9 out of the 11 small-for-size grafts (GRWR < 0.8) in a series of 79 patients who underwent LDLT was higher than 20 mmHg in the early postoperative period (days 2–4) [40]. These patients had poorer survival (84.5% vs. 38.5% at 6 months; $p < 0.01$). They also had increased instances of bacteraemia, cholestasis, ascites and prolonged INR. SAL (performed in seven patients) resulted in a reduction in the median portal pressure from 16 to 11 mmHg, which resulted in improved survival. Interestingly, this was a time when the Kyoto group was still predominantly using right-

lobe grafts (75 in this series) [40]. SAL has shown to decrease PVF from 2600 ± 832 to 1700 ± 689 mL/min with an increase in HAF from 87 ± 39 to 152 ± 64 mL [49]. These authors proposed a portocaval shunt only when the portal flow is >500 mL/min/100 g.

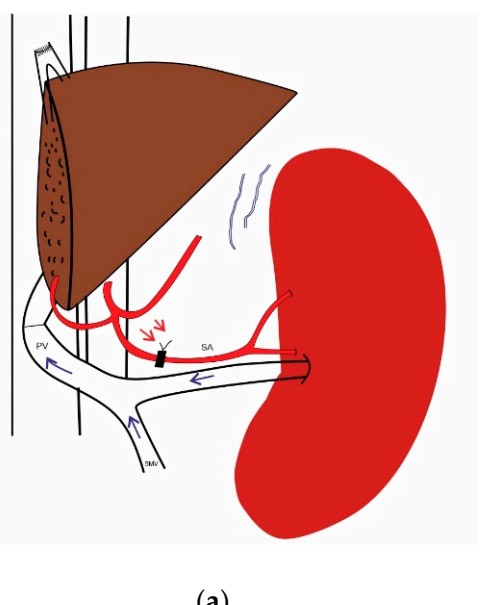

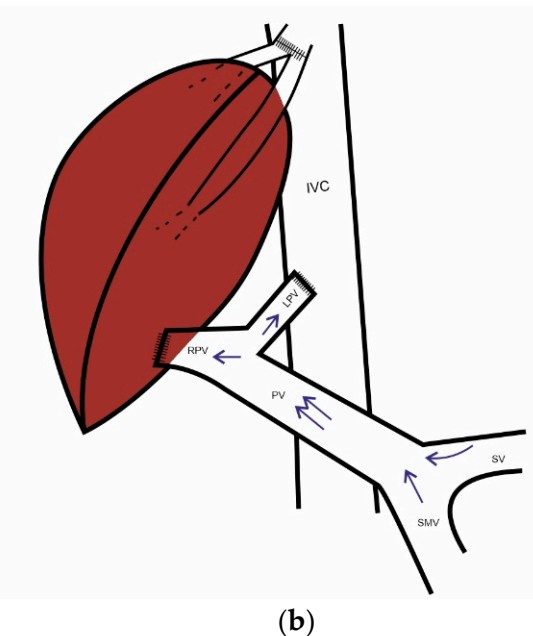

(**a**)                                                                                          (**b**)

**Figure 6.** (**a**) depicts splenic-artery (SA) ligation in the context of left-hemiliver living-donor liver transplantation. This reduces portal inflow by reducing the amount of blood entering the portal vein through the splenic vein. (**b**) The hemiportocaval shunt diverts part of the portal blood into the inferior vena cava, while also maintaining supply to the regenerating partial graft.

### 12.2. Splenectomy

Splenectomy is performed as a graded procedure if SAL does not lower portal pressures below the desired levels, or as a preferred procedure for inflow modulation by certain groups. A retrospective study evaluated 99 patients who underwent adult-to-adult LDLT from June 2014 to December 2020. Of these, 36 underwent GIM, 17 SAL and 19 splenectomy. The latter group had a greater decrease in the postreperfusion PVF [50]. The Kyoto group published their experience of portal-flow modulation based on the time period of the transplant (1998–2006, and 2006–2008) [51]. The use of left-hemiliver grafts increased from 4.9 to 32.1%. Despite the increased usage of small grafts, the one-year patient survival improved from 76.2 to 87.9%. The authors found that in period 2, when the postreperfusion portal pressure was <15 mmHg, 2-year patient survival was better (93% vs. 66.3%) [51]. In a subsequent series, they compared outcomes in grafts with GRWRs > 0.8, between 0.7–0.8 and <0.7 [52]. Authors found that there was no difference in graft or patient survival when portal-inflow modulation was conducted for pressures > 15 mmHg. SFSS was not significantly different (17% vs. 13% vs. 13%), but cholestasis was lower in the group with GRWRs > 0.8. Multivariate analysis demonstrated that donor age >40 years, postreperfusion PVP of >15 mmHg and PVP-CVP > 5 mmHg were significant predictors of poorer patient survival. Propensity-score matching showed that splenectomy resulted in better graft function on postoperative days 7 and 14, lower incidence of SFSS and fewer sepsis episodes at 6 months [34]. Simultaneous splenectomy was recommended when GW/SLV was ≤35 or when postreperfusion PVP > 20 mmHg.

Splenectomy was usually performed in ABO-incompatible liver transplantation prior to the routine use of Rituximab, and used to be common prior to the directly acting antivirals era for hepatitis C infection, when interferons were the mainstay of treatment. Splenectomy in the presence of large collaterals can be a challenging task. Possible complications include intra-/postoperative bleeding, pancreatitis and acute fluid collections, portal vein

thrombosis and the delayed complication of opportunistic postsplenectomy infections by capsulated gram-positive organisms. The use of vessel-sealing devices and vascular staplers might reduce the risk of bleeding and shorten operative time [53]. Splenectomy resulted in better graft survival (90.6% vs. 81.8%) and an acceptable overall complication rate of 10% (pancreatic collections, infections) [53]. Portal-vein thrombosis can occur in up to 10% of patients and is believed to be due to the propagation of the splenic-vein thrombus in the postoperative period, in the setting of low portal flow. Early intervention by reoperation and thrombectomy may salvage the graft.

To circumvent these problems associated with splenectomy, Moon et al. from the Asan group have proposed splenic devascularization [54]. It is a safer alternative with less mortality than splenectomy in the presence of portal hypertension. Here, in addition to SAL, the right gastroepiploic and short gastric vessels are divided, leaving the retropancreatic arteries supplying the spleen as its sole supply. It is yet to be widely adopted as a treatment strategy for SFSS.

*12.3. Portocaval Shunt*

Portocaval shunts are slightly more advanced procedures when considering inflow modulation options. The creation of a portosystemic shunt diverts blood way from the liver. Kokoi et al. described a mesocaval shunt after ligation of the SMV [55]. A mesorenal shunt between the IMV and renal/adrenal vein is another option, described by Sato et al. [56]. A hemiportocaval shunt (HPCS), typically created between the right portal vein and inferior vena cava (IVC), diverts away excess portal blood from the graft and shunts it directly into the IVC (Figure 6b). This shunt is also used by many as a temporary shunt to facilitate recipient hepatectomy. The left portal vein can also be anastomosed to the IVC. This usually reduces portal pressures; however, the excess shunting of portal blood away from a regenerating liver result in graft atrophy and failure. It is difficult to regulate the diameter of the shunt. Many experienced centers use a ligature or rubber band around the portocaval shunt to regulate portal flow in an effort to strike the balance between too little flow, hampering regeneration, and too much flow, causing SFSS. If even the shunt does not lower portal pressure, patients have a poorer outcome. In a large series of 1321 consecutive adult LDLTs over a six-year period, 287 (21.7%) had small grafts with a GRWR < 0.80 [57]. In this group, GIM was performed in 42.9%, HPCS in 109 and SAL in 14 patients. Only 2.8% of patients developed SFSS. Shunt closure by interventional radiology was required in three patients—two in the first month post-transplant and one at five years. Authors concluded that the use of GIM in the form of HPCS and SAL in select patients with a GRWR < 0.8 resulted in good outcomes. HPCS in left-hemiliver grafts, with a GRWR threshold of <0.8 and postreperfusion portal pressure of >20 mmHg has resulted in the avoidance of SFSS [58]. Good results with smaller grafts and inflow modulation reflect maturity in selection polices in these experienced programs. The decision process has to result in choosing a graft that is sufficiently large to meet the metabolic demands of the recipient and achieve the switch towards regeneration rather than failure in the face of portal hyperperfusion; this can be aided by intraoperative portal pressure and flow measurements followed by inflow modulation. Size is a preoperative parameter, flow is an intraoperative parameter and flow modulation becomes important in a smaller graft, which is why most modulation procedures are performed in combination with left-hemiliver grafts.

*12.4. Other Strategies*

Intra operative cineportogram and ligation of shunts: Sometimes, after portal anastomosis, the portal flow may be sluggish even in the absence of thrombosis. This may be due to a steal phenomenon where large perigastric, lineorenal or retroperitoneal collaterals drain into the systemic circulation in a heterotrophic location. If, upon intraoperative clamping, the portal flow and pressures increase to a desirable level, then these can be ligated. The Asan group described the use of an intraoperative cineportogram to accurately identify these collaterals, some of which may be difficult or hazardous to approach surgi-

cally, and provide a real-time measure of improvement in portal flow intraoperatively on their occlusion [54]. The coronary vein provides inflow to the portal vein. Clamping or test occlusion of this vein can be conducted as an option for inflow modulation to assess changes in portal pressure post reperfusion. In many situations, the exact effect of collateral ligation is not apparent until it is finished, and a combination of collateral ligation and portal GIM may be necessary [59].

Splenic-artery embolization (SAE) has been described as an option for inflow modulation. Preoperative proximal SAE performed the evening prior to transplant can reduce intraoperative blood loss and postoperative ascites formation. The mortality rate in patients who received SAE was 3.3% as compared to 13.3% in the non-SAE group, perhaps reflecting the benefit of reduced portal flow. It can be performed in the postoperative period in patients with persistent ascites [60]. However, it has not found wide favor, as sinusoidal injury and the cascade of events that follows thereafter starts almost immediately after reperfusion, and perhaps intraoperative flow modulation is a better option [61].

### 13. Type and Timing of Inflow Modulation

There are no guidelines to choose which form of GIM is best for a particular patient. It depends on author preference, experience and the degree of portal hyperperfusion. SAL/splenic devascularization are safe options. Splenectomy can be performed if it is institutional protocol for GIM and local conditions are not hostile. Japanese centers [52,53] and the recent paper from Cleveland Clinic consider splenectomy their 'go to' procedure for GIM [29]. HPCS can be considered in situations where the GRWR is between 0.6 and 0.7 or postreperfusion portal flow is >500 mL/min/100 g of liver. If portal flow is between 250 and 500 mL/min/100 g, SAL is the simplest procedure to perform. If the postreperfusion PVP decreases from above 20 mmHg to below 20 mmHg, it may suffice. If the pressure does not decrease, a splenectomy can be performed.

Fujiki et al. recommend a prereperfusion splenectomy in recipients with a GRWR < 0.7 and in grafts from donors aged >45 years whose GRWR is between 0.7 and 0.9. They also consider prereperfusion splenectomy in high-MELD (score > 20) recipients whose HVPG is more than 15 mmHg or received grafts with a GRWR < 0.8 [29]. In recipients who received larger grafts (GRWR > 0.9), they performed splenectomy only if postreperfusion hemodynamics were not satisfactory (portal flow > 250 mg/min/100 g, PVP-CVP > 10 mmHg or poor hepatic arterial flow). With this strategy, the authors reported no SFSS, even with the use of small (GRWR < 0.7) grafts. Prereperfusion splenectomy ensures that the graft is not subjected to high portal flow immediately after reperfusion. This prevents early damage that can occur during the time taken for splenectomy if it is carried out post reperfusion. In the paper by Fujiki et al., patients who underwent prereperfusion splenectomy had better outcomes than those who underwent splenectomy after reperfusion [29]. The dissection of the splenic artery prior to recipient hepatectomy saves time if postreperfusion SAL/splenectomy is planned. The test clamp on the splenic artery can be kept on and a ligature can be placed in continuity during splenectomy in the postreperfusion phase to decrease portal inflow to the graft. The release of the portal clamp in a graded manner for reperfusion also serves to prevent the sudden excess flow of portal blood into the graft.

Table 2 summarizes important studies on inflow modulation and its outcomes in liver transplantation [33,35,40,42,45,46,52,57,62–71]. Liver-transplant surgeons have to make several important decisions while choosing the appropriate donor-liver graft for a particular recipient. Real-world situations are not always ideal, as LDLT may be the only option for a high-MELD-score recipient who has high waitlist mortality and poor transplant-free survival. In this context, intraoperative hemodynamic monitoring and GIM offer a 'point of care' intervention strategy that may improve recipient outcomes. General thresholds for GIM using portal flow/pressure/composite indices such as HPi are defined in the literature. Using a modified right-hemiliver graft where the GRWR exceeds 0.8 usually does not require any GIM. Using an SFS graft or left-hemiliver grafts merits serious consideration for GIM. Important caveats include the proper standardization and

measurement of intraoperative hemodynamic parameters, the absence of technical errors such as portal vein anastomotic narrowing/kinking or outflow obstruction. A practical algorithm to manage portal hemodynamics in LDLT is depicted in Figure 7.

**Table 2.** Summary of studies on graft-inflow modulation in living-donor liver transplantation.

| Author/Year (Type of Study) | Number of Patients Type of Graft GRWR | Type of Modulation | Threshold for Modulation Pressure/Flow | Outcome | Comments |
|---|---|---|---|---|---|
| Ito T et al., 2003 [40] (Prospective observational study) | 79 75 right hemiliver two left hemiliver 0.73–2.02% (median, 1.06%) | SAL (7 patients) | Small-for-size graft less than 1.0% of GRWR or PVP $\geq$ 20 mmHg | Cumulative graft survival at 6 months was 83.5%, 86.1% and 38.5% for the SAL, non-SAL low-PVP and non-SAL high-PVP groups, respectively | High postreperfusion portal pressure is associated with poorer survival |
| Trosi R et al., 2003 [42] (Prospective observational study) | 24 23 right hemiliver Mean GRWR 1.12 in patient without GIM, 1.13 in patients with GIM | SAL in 13 patients, one needed additional HPCS | GRWR < 0.8 with PVF > 250 mL/min per 100 g of liver | 3 patients (27%) who did not receive GIM developed SFSS, no patients who received GIM had SFSS 1 YOS was 62% and 93%, respectively, for patients without and with GIM, respectively | GIM modulation prevents SFSS; mean GRWR was >1 in this population |
| Trosi R et al., 2005 [62] (Prospective observational study) | 13 Group 1 without GIM; *n* = 5 (4 right and 1 left hemiliver) GRWR 0.73 (0.58–0.80) Group 2 with GIM; *n* = 8 (equal right, left) GRWR 0.71 (0.56–0.80) | HPCS | GRWR < 0.8 | SFSS 80% without GIM, none with GIM 1-year graft survival was 20% in patients without GIM, 75% with GIM 1-year patient survival was 40% in patients without GIM, 87.5% with GIM | GIM prevented SFSS and improved graft and patient survival |
| Lauro et al., 2007 [63] (Retrospective) | 8 All left hemiliver Two patients had GRWR of 0.4, others had between 0.7 and 0.8 | Splenectomy in 2, splenorenal shunt in 2, splenectomy and portocaval shunt in one | PVP-CVP various thresholds used by authors | SFSS 50% 2 died, 2 retransplanted | Early experience of using left-sided grafts with low GRWR, GIM |
| Yagi S et al., 2008 [45] (Retrospective) | 28 Left-hemiliver graft with caudate in 7, modified right-hemiliver graft in 12 GRWR 0.67–1.60 (median, 1.06%) | Splenectomy (*n* = 4) or splenorenal shunt (*n* = 1) | PVP > 20 | SFSS in 2 patients 1-year graft and patient survival were 92.3% | Early experience of splenectomy as a form of GIM |
| Yoshizumi T et al., 2008 [64] (Retrospective, comparative) | 113 Left hemiliver with caudate (*n* = 63), modified right-hemiliver graft (*n* = 46) GRWR 0.88 $\pm$ 0.20 in patients who did not undergo splenectomy and 0.77 $\pm$ 0.18 in patients who underwent splenectomy | Splenectomy in 44 patients | Portal pressure after portal reperfusion > 20 mmHg | SFSS in 27.4% 4-year patient survival rate in all patients was 85.8%, while that of the without-splenectomy and with-splenectomy groups were 84.4% and 92.1%, respectively | Patients underwent splenectomy for reasons other than GIM as well |
| Ou HY et al., 2010 [65] (Retrospective) | 138 GRWR 1.14 (0.73–1.71) | 6 patients had SAL and one patient also had splenectomy | PVF > 250 mL/min/100 g | 3 out of 8 patients who had PVF > 250 mL/min/100 g developed SFSS; only one with GIM developed SFSS. One patient died | Small number underwent GIM; median GRWR in this study was >1; PVF was the trigger for GIM |
| Ogura et al., 2010 [51] (Retrospective comparative) | 566 502 right, 64 left hemiliver 1.15 in era without GIM, 0.92 during the era of GIM | Splenectomy 84 SAL 1 Portosystemic shunt (IMV-LRV) in addition to splenectomy | GRWR < 0.8 | 12.9% (4 of 31 SFS grafts with a GRWR < 0.8%) developed SFSS Overall, 1-, 3- and 5-year survival rates after LDLT in period I were 76.2%, 71.1% and 68.8%, respectively 1- and 2-year survival rates in period II were 87.9% and 81.6%, respectively | GIM helps in selection of grafts with lower GRWR with similar outcomes as larger grafts without GIM |

**Table 2.** *Cont.*

| Author/Year (Type of Study) | Number of Patients Type of Graft GRWR | Type of Modulation | Threshold for Modulation Pressure/Flow | Outcome | Comments |
|---|---|---|---|---|---|
| Wang et al., 2014 [66] (Retrospective comparative) | 276 Left-hemiliver grafts (*n* = 168, 60.9%) Right-hemiliver grafts (*n* = 108, 39.1%) Mean GV/SLV was 41.8 ± 8.5 | Splenectomy 154 | PVP ≥ 20 mmHg | Incidence of primary graft dysfunction was 9.7% in the splenectomy group and 19.7% (*p* = 0.018) in the non-splenectomy group. 30 patients had early graft loss in 6 months | Splenectomy with PVP > 20 mmHg resulted in better outcomes |
| Osman A et al., 2016 [71] (Retrospective) | 76 1.06 ± 0.22 in patients with PVP less than 15 and 1.00 ± 0.17 in patients with PVP 15–19 | Splenectomy | PVP > 20 mmHg | 6 patients had SFSS when PVP was 15–19 (16.2%) compared to 1 patient (2.6%) in group with PVP < 15 mmHg 9 patients died in group with PVP between 15 and 19 (24.3%), 4 of whom died of SFSS, compared to 3 in the group where PVP was <15 mmHg (7.7%) | The authors proposed 15 mmHg as a cut-off for GIM |
| Uemura T et al., 2016 [52] (Retrospective comparative) | 221 LL 106, RL 115 Average GRWR 0.620 ± 0.0465 (for small), 0.744 ± 0.027 (for medium), 1.010 ± 0.178 (for large) | Splenectomy | Portal pressure > 15 mmHg | 28% SFSS For patients with GRWR < 0.7, 80% of grafts survived at 5 years 49 patients died by 1 year, out of which 14 died of SFSS Satisfactory outcomes in LDLT with GRWR as low as 0.6% using PVP modulation | Authors attributed mortality to nutritional depletion and sarcopenia in low-GRWR patients |
| Emond J et al., 2017 [67] (Multicentric prospective observational study) | 274 233 (85.0%) right hemiliver, 40 (14.6%) left hemiliver and 1 (0.5%) left lateral section GRWR 1.030 (no GIM) GRWR 0.828 (GIM used) | Portocaval shunt 26.9% Splenectomy 8 (15.3%) SAL 34 (65.4%) | Elevated portal pressure reported in 56% of cases, elevated portal flow in 42%. Portal gradient (21%), graft size (15%) and decreased arterial flow (8%) | Graft dysfunction was most common in the SAL patients (42%), two patients of portocaval shunt (17%) and occurred in a single splenectomy patient (13%). Survival at 2 years posttransplant was 90% for the modulated subjects and 81% for the unmodulated subjects | A higher percentage of the modulated (sicker) patients experienced graft dysfunction compared to unmodulated subjects (31% vs. 18%, *p* = 0.03) A2ALL study: variability in practice among participating centers |
| Ito et al., 2016 [68] (Prospective comparative study) | 395 241RL, 154 LL | Splenectomy in 169 | Threshold for modulation was not specified | 5% SFSS The 1-, 3- and 5-year graft survival rates with splenectomy were 88.7%, 85.2% and 81.3%, respectively, and 92.9%, 88.4% and 86.0%, respectively, without splenectomy | Splenectomy in majority of the patients was performed for indications other than GIM |
| Yao S et al., 2018 [33] (Retrospective comparative) | 319 184 RL, 135 LL GRWR < 0.8% in 98 patients (30.7%) | Splenectomy in 59.9% patients, SAL 1, HPCS 1 | Portal pressure > 15 mmHg | Cumulative graft survival was 84.1% in patients needing modulation at 1 year and 75.6% at 5 years. In-hospital mortality was 17.2% | GIM modulation was recommended with GRWR < 0.8 when donors were >45 y/ABO–incompatible |
| Gyoten K et al., 2016 [35] (Retrospective) | 73 Left hemiliver 27 Right hemiliver 45 Estimated GRWR 0.882 (0.464–1.291) in 55 patients where data were available | Splenectomy | PVP > 20 mmHg | 2 patients had SFSS 1-, 3- and 5-year cumulative survival rates were 79.6%, 73.3% and 71.2%, respectively, in the 54 recipients with PVP < 20 mmHg and 89.5%, 77.5% and 69.8% in the 19 with PVP > 20 mmHg followed by splenectomy | Splenectomy with PVP > 20 mmHg results in similar survival to patients with lower pressures, and seems to mitigate effects of low GRWR |

**Table 2.** *Cont.*

| Author/Year (Type of Study) | Number of Patients Type of Graft GRWR | Type of Modulation | Threshold for Modulation Pressure/Flow | Outcome | Comments |
|---|---|---|---|---|---|
| Soin A et al., 2019 [57] (Retrospective comparative) | 1321, out of which 287 had GRWR < 0.8 13 LL 1308 RL GRWR 0.54–0.69 in 79 (5.9%) GRWR 0.70 to 0.74 in 81 (6.1%) GRWR 0.75 to 0.79 in 134 (10.1%) | 109-HPCS, 14-SAL | GRWR < 0.8 No GIM if PVP < 16 PVP 16–18 SAL PVP > 18 HPCS | 2.8% SFSS 0.5% 30-day mortality | Excellent results are a reflection of good selection policy for GIM in an experienced center |
| Miyagi S et al., 2020 [46] (Retrospective) | 188 (83 adults; 105 pediatric) GRWR < 0.8 (*n* = 22) GRWR 0.8–3.5 (*n* = 154) GRWR > 3.5 (*n* = 12) | Splenectomy in 7 patients | GRWR < 0.8 (PVP > 17 mmHg in the later part of the study) | 11.7% SFSS 5 YSR in SFSG without GIM 52.8% 5YSR in SFSG with splenectomy 80.0% | Splenectomy for GIM resulted in similar survival with smaller grafts as with larger grafts without GIM |
| Wong TC et al., 2021 [69] (Retrospective Comparative) | 545 GRWR < 0.6 (*n* = 39; LL 33.3%) GRWR 0.6–0.8 (*n* = 159; LL 10.7%) GRWR > 0.8 (*n* = 347; LL 2.9%) | | GRWR < 0.6 | 4.7% SFSS 2 in-hospital mortalities | Good results with GIM despite the use of smaller grafts |
| Hye-Sung Jo et al., 2022 [70] (Retrospective comparative) | 118 93 RL, 25 LL | SAL | Portal flow >300 mL/min/100 g and HVPG > 10 mmHg | SFSS 16% in LL and 3.2% in RL No SFSS-related mortalities | SAL based on intraoperative portal flow and HVPG results in satisfactory outcomes |

SAL—Splenic-artery ligation; GRWR—Graft-recipient-weight ratio; PVP—Portal venous pressure; CVP—Central venous pressure; GIM—Graft-inflow modulation; HPCS—Hemiportocaval shunt; OS—Overall survival; SFSS—Small-for-size syndrome; PVF—Portal vein flow; IMV—Inferior mesenteric vein; LRV—Left renal vein; LDLT—Living-donor liver transplantation; GV—Graft volume; SLV—Standard liver volume; RL—Right lobe; LL—Left lobe; HVPG—Hepatic-vein pressure gradient.

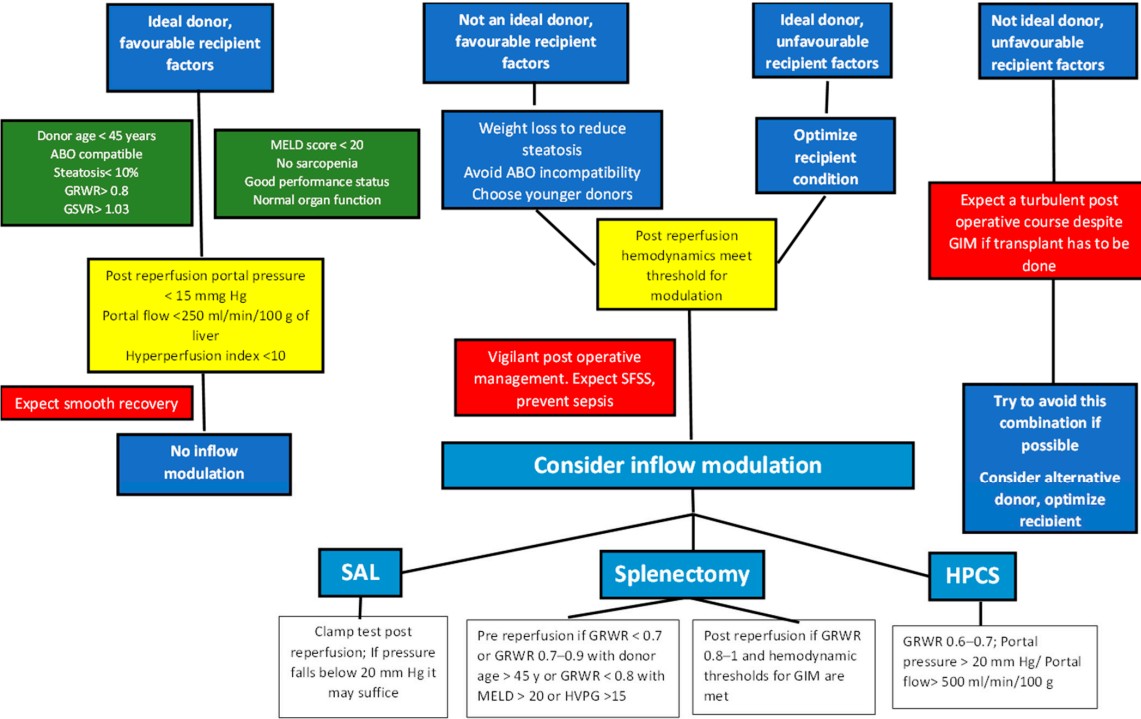

**Figure 7.** A practical algorithm to depict the role of inflow modulation in the context of living–donor liver transplantation. GRWR—graft–recipient–weight ratio; GSVR—graft–spleen–volume ratio; MELD—model for end–stage liver disease; GIM—graft–inflow modulation; SAL—splenic–artery ligation; HPCS—hemiportocaval shunt. Recommendations (white boxes) for SAL are adapted based on Ito et al. [37] for splenectomy are based on Fujiki et al. [29] and for HPCS based on Yamada et al. [58].

## 14. Pharmacological Measures

An increase in hepatic resistance contributes to portal hypertension as cirrhosis progresses. At a later date, an increase in portal flow sustains portal hypertension. Initial clinical hypotheses of the mechanisms of portal hypertension were tested in animal models, and the molecular factors involved were elucidated [72]. Pharmacological measures that reduce portal pressure but preserve HABR have a favorable effect on graft function. Mehrabi et al. investigated the role of systemically administered vasopressors (epinephrine and norepinephrine) and found that both reduced PVF and hepatic arterial flow in porcine liver transplantation [73]. Vasopressin and Terlipressin have a selective effect on portal pressure due to their action on V1a receptors [74]. Wagener et al. [75], in a study of 16 patients, found that Vasopressin decreased portal pressure and flow by causing splanchnic vasoconstriction, without affecting cardiac output or mean arterial pressure. In a double-blind randomized controlled trial of the routine perioperative use of Terlipressin in adult LDLT by Reddy MS et al., authors did not find any reduction in postreperfusion portal pressure with the systemic use of Terlipressin [76]. However, Terlipressin infusion reduced ascites formation and, therefore, the need for paracentesis. The length of hospital stay was lower in the Terlipressin group. Due to possibility of side effects such as a rise in lactate level and symptomatic bradycardia, Terlipressin should be used with close monitoring in patients with high-volume ascites. In a rat-liver transplantation model, the use of low-dose Somatostatin reduced graft injury, postulated to be due to a reduction in shear stress due to increased portal flow [77]. Granulocyte-colony-stimulating factor [78] and hyperbaric oxygen treatment [79] have been shown to reduce liver injury in a massive hepatectomy model in rats. In study by Suehiro T et al., authors continuously administered intraportal Nafamostat mesilate, a Prostaglandin E1 and Thromboxane A2 synthetase inhibitor, by double-lumen catheters for 7 days [80]. This reduced hyperbilirubinemia and ascites in patients with small-for-size grafts. There are also case reports in which propranolol, alone or along with somatostatin, was used to treat SFSS that occurred despite GIM [81,82]. A randomized trial by Troisi et al. found that somatostatin infusion reduced HVPG while preserving arterial inflow to the graft [83]. Pharmacological modulation has the potential to augment the effects of surgical modulation; this needs further study. Interventions to improve hepatic artery flow in small-for-size livers is another strategy. Kelly DM, Zhu X et al. demonstrated improved survival with the infusion of adenosine into the hepatic artery in an animal model [84]. Currently, these methods are not used widely.

## 15. Future Directions

Current GIM strategies follow a one-size-fits-all threshold for pressure or flow. Additionally, it is not clear which method of GIM is best, and it is not entirely predictable how much the portal pressure or flow will change with GIM. Predicting how postreperfusion portal pressure and flow will behave in a particular donor-liver-recipient hemodynamic bed with GIM perhaps requires complex modeling and an equation derived by artificial intelligence fed with multiple data points. Additionally, future strategies for GIM can be engineered to be personalized and dynamic so that the liver is able to autoregulate its regeneration and sinusoidal pressure to reach equilibrium with the recipient portal bed in the shortest possible time. This requires advances in technology and molecular medicine to influence portal hemodynamics. Regenerative preconditioning of the graft liver is an interesting strategy that may improve graft regeneration [85]. Genetically modified or enhanced grafts may tolerate portal hyperperfusion better and be less prone to allograft dysfunction.

**Author Contributions:** Concept and plan: K.G.B. Manuscript preparation: K.G.B. and S.S. Manuscript finalization: K.G.B. and S.S. All authors have read and agreed to the published version of the manuscript.

**Funding:** This research received no external funding.

**Institutional Review Board Statement:** Not applicable.

**Informed Consent Statement:** Not applicable.

**Data Availability Statement:** Not applicable.

**Conflicts of Interest:** The authors declare no conflict of interest.

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
