# Peer review of "Portal Hemodynamics after Living-Donor Liver Transplantation: Management for Optimal Graft and Patient Outcomes—A Narrative Review"

_2673-3943, doi:10.3390/transplantology4020006_

Round 1
Reviewer 1 Report
- The authors should emphasize the evolution of the SFSS definition from a purely clinical syndrome (Dahm et al, AJT 2015) to the inclusion of a hemodynamic parameters in the more recent definitions (see Hernandez-Alejandro et al, Liver Transpl 2016). This shift from the size to the flow paradigm should be emphasized by the authors.
To clarify this point to the reader, the authors should organize their review in accordance. Wile size remains important (GRWR) inflow, outflow and pressure play a major role in LDLT.
Indeed, as recently presented by Fujiki et al in a landmark paper there are three important steps to consider:
1. Reducing the risk for the donor (use left grafts including the MHV and S1) and Matching with the recipient risk
2. Measuring and modulating the portal inflow during DDLT as the SSFS occurs in the early post-operative phase and discuss the different portal flow modulation strategies available as well as their timing
3. Outflow optimization is also a major outcome indicator in partial grafts which is also related to portal flow and HVPG.
These different points are discussed by a recent landmark paper by Fujiki et al (DOI: 10.1097/SLA.0000000000005630) which is not cited by the authors. Fujiki et al also present a very clear decision algorithm which should be incorporated in the algorithm proposed by the authors.
- The description of the different patho-physiological mechanism of chronic liver disease are important but may be reduced to provide a more precise and concise manuscript. We would however suggest to discuss preoperative recipient assessment as well as potential pre-operative indicators which could indicate a risk of SFSS such as presence of porto-systemic shunt. In this context, the authors could discuss if fulminant hepatitis recipient are less at risk to SFSS or if optimal matching is even more important given their high MELD score etc.. (See algorithm of Fujiki et al)
- The authors should also discuss when to perform hemodynamic measurements (prior to explantation, after implantation), the role of performing clamping tests of splenic artery prior to ligation or splecnetcomy.
- Is there a place for partial portal declamping in case of expected high portal flow to prevent early exposure of the graft to high portal flow.
- The hepatic arterial buffer response is an important factor especially in case of a high portal flow. As shown in a large cohort of whole grafts by Matsushima et al (Transplantation 2020), grafts with a high PF present with more hepatic artery thrombosis and biliary complications after LT. This should also be discussed.
- Given the complexity of the subject the authors should provide illustrative clinical cases.
- Please avoid citing entire phrasing from already published papers (for example p. 8 line 6 copied from Piscadlia et al)
Author Response
Thank you for the seasoned comments and suggestions to improve the manuscript. We have addressed each of the points and tried to make changes as advised.
The authors should emphasize the evolution of the SFSS definition from a purely clinical syndrome (Dahm et al, AJT 2015) to the inclusion of a hemodynamic parameters in the more recent definitions (see Hernandez-Alejandro et al, Liver Transpl 2016). This shift from the size to the flow paradigm should be emphasized by the authors.
The evolution of the definition of SFSS from a clinical syndrome to inclusion of hemodynamic parameters with shift in emphasis from purely size to inclusion of flow has been highlighted. In 2015, Dahm et al proposed a definition of SFS dysfunction in a small partial liver graft (GRWR <0.8) as presence of two criteria (bilirubin >100umol/l, INR>2, grade 3/4 encephalopathy) on three consecutive days in the first post operative week after the exclusion of technical (arterial/portal occlusion, outflow congestion, bile leak), immunological (rejection) or infectious (cholangitis, sepsis) causes. Subsequently Hernandez- Alejandro et al proposed a more comprehensive definition including portal flow (>250 ml/min/100 g liver) as a prerequisite in addition to size (GRWR <0.8). They considered SFSS to be present if two of the four parameters (ascites, hyperbilirubinemia, prolonged INR, hepatic encephalopathy) were present (Table 1) in the absence of technical/immunological or infectious causes. The inclusion of portal flow marks the shift in understanding from size alone to size and flow paradigm in the pathogenesis of SFSS.
To clarify this point to the reader, the authors should organize their review in accordance. While size remains important (GRWR) inflow, outflow and pressure play a major role in LDLT.
Indeed, as recently presented by Fujiki et al in a landmark paper there are three important steps to consider:
- Reducing the risk for the donor (use left grafts including the MHV and S1) and Matching with the recipient risk
- Measuring and modulating the portal inflow during DDLT as the SSFS occurs in the early post-operative phase and discuss the different portal flow modulation strategies available as well as their timing
- Outflow optimization is also a major outcome indicator in partial grafts which is also related to portal flow and HVPG.
These different points are discussed by a recent landmark paper by Fujiki et al (DOI: 10.1097/SLA.0000000000005630) which is not cited by the authors. Fujiki et al also present a very clear decision algorithm which should be incorporated in the algorithm proposed by the authors.
Thank you for providing nice points to organize and present our thought processes more clearly. We have referenced the recent landmark paper by Fuijki et al and added new sections to present these points better.
Our main emphasis in the manuscript is to discuss portal hemodynamics in LDLT and how to optimize them for better outcomes. Donor selection, performance of a technically sound operation (inflow as well as outflow reconstruction) are important factors closely inter-related to recipient outcomes.
We had not discussed donor selection as there are excellent papers available on this topic. A paragraph on important factors to consider while selecting a donor and matching with recipient risk is added. A new figure is also included to highlight this.
In the definition of SFSS, technical factors (such as outflow and inflow problems) are excluded. Therefore, our paper assumes a technically sound reconstruction while discussing the effects of portal hyperperfusion on the graft. However, reading the paper by Fuijki et al was insightful, especially the use of all three hepatic veins in the recipient for outflow reconstruction in a left hemiliver graft with caudate. With this strategy, the authors have reported no SFSS even with the usage of small (GRWR< 0.7) grafts.
Also, the liberal usage of splenectomy both pre and post reperfusion for inflow modulation and their algorithm are quoted in the revised version of our manuscript. We have modified our algorithm to present a better understanding of these interrelated factors.
- The description of the different patho-physiological mechanism of chronic liver disease are important but may be reduced to provide a more precise and concise manuscript. We would however suggest to discuss preoperative recipient assessment as well as potential pre-operative indicators which could indicate a risk of SFSS such as presence of porto-systemic shunt. In this context, the authors could discuss if fulminant hepatitis recipient are less at risk to SFSS or if optimal matching is even more important given their high MELD score etc.. (See algorithm of Fujiki et al)
The section on pathophysiological mechanisms of chronic liver disease has been edited for brevity. Recipient factors that may portend SFSS such as high MELD score, presence of splenomegaly are emphasized. While patients with fulminant liver failure may not have significant portal hypertension, they may still require a higher parenchymal mass to meet the metabolic demand of a sick recipient.
- The authors should also discuss when to perform hemodynamic measurements (prior to explantation, after implantation), the role of performing clamping tests of splenic artery prior to ligation or splecnetcomy.
The best single point of time to perform hemodynamic measurements is after implantation, reperfusion. This gives a decision point to perform inflow modulation. If factors predictive of SFSS are present, inflow modulation (splenectomy) can be performed prior to recipient hepatectomy as proposed by Fujiki et al. Hemodynamic measurements in these cases can be performed at three time points (before splenectomy, after splenectomy and after reperfusion).
The role of clamp test of the splenic artery prior to ligation or splenectomy has been highlighted.
- Is there a place for partial portal declamping in case of expected high portal flow to prevent early exposure of the graft to high portal flow.
This is an interesting point which we have not addressed. Releasing the portal clamp in phases may prevent a small partial graft in a hyperdynamic milieu from receiving excess portal blood. In these instances, it is better to have isolated the splenic artery in the lesser sac in the dissection phase prior to recipient hepatectomy. Exposure to high portal flows can be mitigated quickly by ligating the already dissected splenic artery in continuity. This can be followed by a splenectomy if it is the policy for inflow modulation in the treating unit. In the paper by Fujiki et al, patients who underwent pre-reperfusion splenectomy had better outcomes than those who underwent splenectomy after reperfusion.
- The hepatic arterial buffer response is an important factor especially in case of a high portal flow. As shown in a large cohort of whole grafts by Matsushima et al (Transplantation 2020), grafts with a high PF present with more hepatic artery thrombosis and biliary complications after LT. This should also be discussed.
We had alluded to this point briefly in the manuscript. We have discussed the paper by Matsushima et al. to emphasize the higher risk of HAT with high portal flow to the graft. Thank you for the suggestion.
- Given the complexity of the subject the authors should provide illustrative clinical cases.
Clinical cases would have illustrated decision making process more clearly. Unfortunately, we do not have the entire gamut of possible presentations to make an impactful discussion. Apologies for this. We have made the figures and algorithms better so that the reader gets a grasp of principles of portal hemodynamics. We earnestly hope the review will provide broad guidelines and understanding while making decisions.
- Please avoid citing entire phrasing from already published papers (for example p. 8 line 6 copied from Piscadlia et al)
The manuscript has been proof read and these aspects have been rectified. Thank you.
Author Response
Firstly, I like to congratulate the authors for their efforts. The article is well written and even though it does not provide “new” knowledge it serves as a good summary and provides important information to the reader. Overall, I would recommend a thorough prove read of the text as it contains “may”, “could” etc. on multiple occasion which should be replaced if the data supports the statement.
Thank you for the encouraging and valuable comments to improve the manuscript. We have re-read the manuscript and tried to provide support from existing literature for most of the statements.
A detailed explanation of some of my more specific concerns are reported below.
- The Authors focus on living donor liver transplantation (LDLT) and in their introduction describe the "search” for a possible donor. In my understanding the concept of LDLT always has the problem of donor selection as the pool of possible donors is very small. I think it is important to elaborate the differences between living and postmortem donation. Especially on an ethical and immunological aspect. (Line 57-67)
Thank you for raising this important point. Although the main focus of our manuscript is not donor selection in LDLT, it is an important factor that has a bearing on outcomes even in the context of hemodynamic factors. A new section has been added on donor selection and differences between living donor and deceased donors. Preoperative recipient assessment based on metabolic status, portal hemodynamic status and performance status has been discussed and the principles of donor recipient matching in LDLT are briefly described.
- The Author describes the definitions for small for size Syndrome in Table 1. The first 3 Publication appear to be from the same Author Soejima Y et al., to the reader it is difficult to follow weather he revised his own definitions or weather he just added further information. Please comment and try to make it easier to follow.
Table 1 has been changed to provide clarity on the definitions for the reader. In the first publication by Soejima et al (2003), authors defined SFSS using 2 parameters bilirubin and ascites. Subsequently in 2006, they have increased the level of bilirubin from 5 to 10 mg/dl in the definition of SFSS. The paper by Dahm et al provides distinction between small for size dysfunction and non-function/failure. The errors from the previous table have been rectified.
- In Line 362-364 the authors recommend splenectomy when a small graft is predicted preoperatively. Is there any data supporting this statement? The Authors describe the challenges which arise when splenectomy is necessary, therefore it appears even more important that this recommendation is based on morbidity/mortality data rather than personal experience.
Yes, this is an important point. Decision for splenectomy should be weighed carefully in light of potential complications. The Kyoto (Ref 52) and Kyushu (Ref 53) groups recommend splenectomy for inflow modulation. The recent paper by Fujiki et al (Ref 29) also advocated splenectomy when small grafts are used in the context of portal hyperperfusion. All these points are discussed in the revised version of the manuscript.
- In the current format Table 2 is unreadable for Review. Please rearrange so it can be further evaluated.
Thank you for this suggestion. Table 2 has been rearranged for better readability
- Figure 6 is a good idea, and I think it is a good summary for the reader. As a suggestion I would rather use square boxes as for example the BCLC uses. This makes it much easier to read and appears more organized. (https://www.sciencedirect.com/science/article/pii/S0168827821022236)
The boxes have been changed to square ones as advised
After remodulation I would add one more row below “Consider inflow modulation” and describe the different inflow modulation as this is the main point of the manuscript.
The algorithm has been modified to make it more comprehensive and reflect the decision processes involved. Thank you.
Round 2
Reviewer 1 Report
We thank the authors for providing a revised version of the paper which is now sustantially improved. Please find some minor comments below: Please also include in Table 1 the fact that Hernandez-Alejandro et al also included portal flow as a criteria for SFSS. The subtitle "Sound surgical technique" is a somewhat misleading term as every technique should be sound. Please modify this title. Please correct the typo in the sentence: "The best single 386 point of time to perform hemodynamic measurements is after implantation, reperfusion."
Author Response
The authors thank the reviewers for their time, kind comments and suggestions that have improved the quality of the manuscript.
Table 1 has been modified to reflect the fact that Hernandez-Alejandro et al also included portal flow > 250ml/min/100 g in addition to GRWR < 0.8 as a criteria for SFSS.
The subtitle "Sound surgical technique" has been changed to "Outflow reconstruction".
The sentence with the typo has been modified
Reviewer 2 Report
Thank your for the extensive Review. My issues have been adressed.
Author Response
Thank you for the time and valuable suggestions to improve the quality of manuscript